# TimeTector: A Twin-Branch Approach for Unsupervised Anomaly Detection in Livestock Sensor Noisy Data (TT-TBAD)

**DOI:** 10.3390/s24082453

**Published:** 2024-04-11

**Authors:** Junaid Khan Kakar, Shahid Hussain, Sang Cheol Kim, Hyongsuk Kim

**Affiliations:** 1Department of Electronics and Information Engineering, Jeonbuk National University, Jeonju 54896, Republic of Korea; junaidk@jbnu.ac.kr; 2Core Research Institute of Intelligent Robots, Jeonbuk National University, Jeonju 54896, Republic of Korea; sckim7777@gmail.com; 3Innovation Value Institute (IVI), School of Business, National University of Ireland Maynooth (NUIM), W23 F2H6 Maynooth, Ireland; shahid.hussain@mu.ie; 4Department of Electronics Engineering, Jeonbuk National University, Jeonju 54896, Republic of Korea

**Keywords:** anomaly detection, unsupervised learning, time series sensor, LSTM autoencoder, multi-head attention mechanism, DAE, reconstruction and prediction branch, livestock

## Abstract

Unsupervised anomaly detection in multivariate time series sensor data is a complex task with diverse applications in different domains such as livestock farming and agriculture (LF&A), the Internet of Things (IoT), and human activity recognition (HAR). Advanced machine learning techniques are necessary to detect multi-sensor time series data anomalies. The primary focus of this research is to develop state-of-the-art machine learning methods for detecting anomalies in multi-sensor data. Time series sensors frequently produce multi-sensor data with anomalies, which makes it difficult to establish standard patterns that can capture spatial and temporal correlations. Our innovative approach enables the accurate identification of normal, abnormal, and noisy patterns, thus minimizing the risk of misinterpreting models when dealing with mixed noisy data during training. This can potentially result in the model deriving incorrect conclusions. To address these challenges, we propose a novel approach called “TimeTector-Twin-Branch Shared LSTM Autoencoder” which incorporates several Multi-Head Attention mechanisms. Additionally, our system now incorporates the Twin-Branch method which facilitates the simultaneous execution of multiple tasks, such as data reconstruction and prediction error, allowing for efficient multi-task learning. We also compare our proposed model to several benchmark anomaly detection models using our dataset, and the results show less error (MSE, MAE, and RMSE) in reconstruction and higher accuracy scores (precision, recall, and F1) against the baseline models, demonstrating that our approach outperforms these existing models.

## 1. Introduction

Anomaly detection has been a central focus in machine learning research for many years. An anomaly in time series data is a deviation from the expected pattern or trend within a given time frame, showing up as outliers, fluctuations, or changes in the data, indicating unusual behavior [1]. Livestock anomalies refer to unexpected changes in trends or patterns, such as irregularities in population dynamics, production metrics, or environmental factors like temperature, humidity, and gases. It has widespread applications across various domains, including livestock farming [2], cyber-intrusion detection, healthcare, sensor networks, and image anomaly detection. The expansion of livestock farms and the number of animals necessitate advanced management techniques for optimizing productivity with limited manpower. Failure to address the irregularities in livestock data can lead to various consequences, including untreated illnesses, reduced productivity, and heightened mortality rates, posing significant risks to animal welfare and farm profitability [3]. Detection and management of these anomalies are crucial for ensuring optimal farm management and animal well-being. Precision farming emerges as a solution, propelling the transition toward automated livestock smart farms [4]. Central to this is anomaly detection, which is crucial in sectors like finance and healthcare but pivotal in livestock management. Farmers can gain valuable insights by detecting outliers in data points that deviate significantly from typical patterns, which are frequently caused by discrepancies and errors in the sensors. Organizations are exploring methods to enhance pig productivity with minimal human involvement. Livestock health, well-being, and productivity can be significantly improved by paying close attention to them and promptly detecting anomalies. This can help in identifying potential health issues or environmental inconsistencies in livestock settings, allowing for swift interventions to maintain animal welfare and farm productivity. Studies have demonstrated that focusing on the livestock sector can result in positive outcomes, as evidenced by a wealth of research [4,5]. Given the increasing scale of livestock rearing, adopting intelligent agricultural solutions is essential to ensuring optimal farm and livestock housing conditions, especially in pig farming, where it is crucial. The seamless integration of different equipment within and around barns is essential for creating and maintaining optimal growth conditions for livestock. Gas sensors in pig barns ensure the safety and health of animals and workers. Conventional outlier detection techniques lack efficacy when dealing with large, multi-dimensional datasets. Autoencoders, representing a type of deep learning technique, have excelled in detecting outliers due to their ability to identify complex patterns and relationships in the data. Autoencoders have revolutionized anomaly detection and prediction [6]. Advanced analytical techniques have the capability to identify complex, nonlinear relationships within data. These techniques allow for the precise identification of anomalies in offering predictive insights essential for improving the efficiency and productivity of livestock management.

Having accurate data is essential for making informed decisions. Several factors can cause variations in data quality, including measurement errors, inaccurate data entries, sensor failures, or deliberate sabotage. These challenges can be complex to manage using sophisticated methodologies. Analyzing spatial interconnections within multi-sensor datasets is crucial for making informed decisions in the current era. The complexity lies in differentiating between noise, typical, and atypical data. Therefore, there is a need for a dependable mechanism that enables precise identification and interpretation of such data. Several approaches for analyzing spatial dependencies in multiple sensors have been studied [7,8]. To deal with such challenges, researchers have developed various methods, e.g., data-driven anomaly detection [9] and maximum mean discrepancy (MMD) [10]. Deep learning approaches, especially those involving autoencoders, have surfaced as potent solutions for outlier detection, although some may fall short in performance. Their performance is often tied to their adeptness in unraveling intricate nonlinear relationships within data [11,12]. Through learning intricate data patterns, these techniques facilitate enhanced anomaly identification, fostering a more accurate and insightful analysis.

Pig house sensors collect real-time management data, ensuring optimal growth and animal welfare. Equipment inside and outside the house maintains a suitable environment and boosts productivity. Gas sensors monitor hazardous gases like carbon dioxide CO_2_ and ammonia NH_3_ which can accumulate due to animal respiration and waste. Integrating gas sensors with ventilation systems helps control the indoor climate and maintain a healthy environment for animals and workers. High carbon dioxide CO_2_ levels can pose a suffocation risk [13,14]. Using mechanisms that capture spatial–temporal correlations and dependencies among diverse data streams has proven highly effective. Traditional models like RNNs, VAEs, or GANs may need to become more capable of addressing this challenge with the same level of efficiency [15]. Using our proprietary dataset from the pig farm, we show that pig diseases are the most critical factor affecting productivity. Preventing infections is crucial for maintaining pig productivity. Proper maintenance of the pig house and equipment can create a healthier environment and decrease the chances of infection [16,17].

Autoencoder-based models are effective in smart farming, especially in anomaly detection. A study found that they can detect anomalies in agricultural vehicles by identifying obstructive objects in the field. The study also introduced a semi-supervised autoencoder trained with a max-margin-inspired loss function that outperformed basic and denoising autoencoders in the generation of anomaly maps using relative-perceptual–L1 loss [18]. According to the research, autoencoders are identified as a versatile tool for anomaly detection. Autoencoders use an encoder and a decoder to minimize reconstruction errors for normal data. Anomalies result in higher reconstruction errors and are identifiable as outliers [19,20].

Several techniques have been proposed to overcome the limitations of outlier detection: LSTM, LSTM-AE, ConvLSTM, and GRU. These standard techniques have limitations in identifying the capacity to detect a wider range of subtle and complex anomalies, especially in intricate, dynamic industrial environments [21,22]. One such technique is represented by ensemble methods which combine the outputs of multiple models to improve accuracy and robustness. Like in any technology, there is always a chance of malfunction, probably leading to undetected hazardous conditions [23]. Advanced analytics and machine learning improve the speed and accuracy of real-time data processing, enabling precise anomaly detection and interpretation of complex data relationships in livestock farming. However, the growing use of IoT devices in agriculture raises security and privacy concerns that must be addressed. Current anomaly detection methods in agriculture, especially in livestock farming and industrial settings, must handle real-time, multi-dimensional data. Traditional autoencoder architectures struggle to adapt to the complexities of dynamic and noisy farm environments. Unpredictable variables often affect data in this field, resulting in a higher rate of false positives or negatives.

In our research, we present a new model called the “TimeTector (TT-TBAD)”, which offers a revolutionary approach to detecting anomalies in agricultural environments. Our innovative methodology is specifically designed to handle the complexities and nuances of multi-sensor and real-time series data that are common in farming settings, thereby providing a robust and advanced autoencoder ensemble approach that addresses the unique challenges of monitoring pig farm environments. It is capable of detecting outliers with remarkable accuracy and can also make predictive inferences, which is a feature that is often absent in traditional models. As a result, “TimeTector (TT-TBAD)” represents a significant advancement in the field, providing an effective and precise solution for detecting anomalies in critical agricultural applications. Our innovative TT-TBAD methodology is designed to detect anomalies in livestock sensor data using advanced techniques. It utilizes a Twin-Branch Shared LSTM Autoencoder framework, incorporating a Denoising Autoencoder (DAE) and several Multi-Head Attention mechanisms. This approach can be used to effectively handle the noisy data commonly found in agricultural settings. The model captures complex spatial–temporal correlations and hierarchical relationships while minimizing reconstruction and prediction errors. Real-time performance insights are provided via a dynamic feedback loop. Optimized for multi-sensor time series data, TT-TBAD enhances data quality by reducing noise and providing accurate forecasts, enabling proactive farm management and improved productivity and animal welfare. In summary, this study makes three key contributions:This model integrates advanced mechanisms, including Multi-Head Attention, Hierarchical Multi-Head Attention, Cross Multi-Head Attention, and Feature-Enhanced Multi-Head Attention. These features enable the analysis of hierarchical data, identification of complex relationships, and interpretation of auxiliary features. Moreover, the model’s dual-branch LSTM structure enhances spatial–temporal data analysis, resulting in improved accuracy for anomaly detection in livestock farming, a capability not adequately addressed by traditional models.The integration of a Denoising Autoencoder (DAE) with Shared LSTM encoder branches manages noisy agricultural sensor data during training, mitigating overfitting and enhancing adaptability. The development of a Twin-Branch structure for multi-task learning enables simultaneous data reconstruction and future prediction, distinguishing between normal operations, noisy data, and anomalies. This dual-task learning approach generates precise, actionable insights in real time, a capability not typically found in traditional models.The developed model uses a feedback loop based on reconstruction and prediction errors to continuously monitor and adjust its performance, through a dynamic rather than static threshold-based mechanism. This adaptive mechanism significantly improves the detection accuracy and the model’s adaptability to evolving farm conditions by boosting the accuracy and adaptability in anomaly detection, aiding decision-making. Finally, evaluation results showed a lower construction on error (MSE, MAE, and RMSE) and high accuracy (precision, recall, and F1) when evaluated against the benchmark and baseline anomaly detection models.

The rest of the paper is organized as follows: Section 2 provides related work, exploring state-of-the-art techniques for anomaly detection in sensor data. Section 3 and Section 4 describe our proposed methodology and detailed framework. The dataset explanation and analysis are described in Section 5. Performance evaluation, results, and analysis of the experiment follow in Section 6. Finally, Section 8 concludes the paper with directions for possible future work.

## 2. Related Work

Several research studies have been performed on existing state-of-the-art techniques for detecting anomalies in indoor and outdoor pig farm environment datasets and other similar fields [24]. This detection process has been studied for decades in natural language processing (NLP) and images and videos [25]. Various methods are available for anomaly detection, including statistical, supervised machine learning-based, and unsupervised deep learning-based approaches. These methods have been explored in detail through literature reviews in similar studies; as such, our focus is on unsupervised learning due to the lack of labels. Outlier detection in real-time series data has unique challenges that require efficient processing and handling of data drift or concept drift [26]. Machine learning and deep learning advancements in analyzing time series data from sensor networks show that spatial information from sensor locations can be leveraged to improve predictions [27].

The use of techniques such as sliding windows, feature selection, and online learning algorithms for handling real-time series data in outlier detection tasks has been explored in previous research, where challenges specific to real-time series data in outlier detection have been acknowledged [28]. These challenges include handling the data drift, where the statistical properties of the data change over time, and the concept drift, where there is the evolution of the underlying concepts being monitored. To address these challenges, techniques like sliding windows, which limit the data window for analysis, and feature selection, which focuses on the most relevant attributes, have been explored [29]. Additionally, online learning and multilayer neurocontrol of high-order uncertain nonlinear systems with active disturbance rejection algorithms, which adapt to changing data distributions and solve real-world problems, have been investigated to improve the efficiency of outlier detection in real-time scenarios [30,31].

Additionally, autoencoder-based anomaly detection has proven to be highly effective in IoT-based smart farming systems, exhibiting exceptional performance in accurately detecting anomalies in key variables such as soil moisture, air temperature, and air humidity. The integration of this technology has shown superior performance, making it a valuable tool for precision agriculture. This study proposes a new approach for anomaly detection using autoencoder-based reconstructed inputs. The proposed method outperforms existing techniques and facilitates the creation of supervised datasets for further research. The hourly forecasting models for soil moisture and air temperature demonstrate low numbers of estimation errors, indicating the effectiveness of this method [32].

In order to improve the accuracy of intrusion detection on the Internet of Things (IoT), intrusion labels are integrated into the decoding layers of the model. However, this approach may not effectively handle the data and concept drift challenges frequently encountered in real-time anomaly detection. Additionally, multi-sensor technology IoT devices can be used in both cloud and edge computing infrastructure to provide remote control of watering and fertilization, real-time monitoring of farm conditions, and solutions for more sustainable practices. However, these improvements in efficiency and ease of use come with increased risks to security and privacy. Combining vulnerable IoT devices with the critical infrastructure of the agriculture domain broadens the attack surface for adversaries, making it easier for them to launch cyber-attacks [33]. An end-to-end deep learning framework for extracting fault-sensitive features from multi-sensor time series data showcases data-driven methods to handle spatiotemporal cases [34].

Anomaly detection involves a comprehensive survey of deep learning techniques for anomaly detection, covering various methods and their applications across different domains. However, the survey might not delve deep into real-time series data anomaly detection [35]. A survey on anomaly detection that focused on time series using deep learning and covered a range of applications, including healthcare and manufacturing, provided specific insights into livestock health monitoring or multi-task learning [36]. Knowledge distillation for compressing an unsupervised anomaly detection model and addressing the challenge of deploying models on resource-constrained devices are examples of the challenges associated with real-time data processing and multi-task learning in anomaly detection [37]. One learning-based study introduced a simple yet effective method for anomaly detection by learning small perturbations to perturb normal data and subsequently classifying the normal and perturbed data into two different classes using deep neural networks. The focus on perturbation learning might not align with the Multi-Head Attention and memory-driven multi-task learning approaches in our work [38].

## 3. Preliminaries

### 3.1. LSTM Network

Hochreiter and Schmid Huber introduced LSTM in 1997. It is an advanced RNN that addresses the vanishing gradient problem and models long-term dependencies; Figure 1. LSTM is popular in commercial applications for its memory cells controlled by gate units [39], including input, forget, and output gates, to selectively store information over time as given in Equation (Equation 1). A detailed explanation is given in Section 4.
(1)$$\begin{array}{c} LSTM_{Structure}=\left\{\begin{array}{cc} f_{t} & =\sigma (W_{f}\times \left[h_{t-1},x_{t}\right]+b_{f})\\ i_{t} & =\sigma (W_{i}\times \left[h_{t-1},x_{t}\right]+b_{i})\\ {\tilde{c}}_{t} & =\tanh(W_{c}\times \left[h_{t-1},x_{t}\right]+b_{c})\\ c_{t} & =f_{t}\times c_{t-1}+i_{t}\times {\tilde{c}}_{t}\\ o_{t} & =\sigma (W_{o}\times \left[h_{t-1},x_{t}\right]+b_{o})\\ h_{t} & =o_{t}\times \tanh\left(c_{t}\right) \end{array}\right. \end{array}$$

### 3.2. Encoder–Decoder

Autoencoders use an encoder to compress input data into a “latent space” and a decoder to reconstruct original data as shown in Figure 2. Minimizing the reconstruction error during training allows for effective feature extraction and data reconstruction, presented by Equation (Equation 2) [21,33,34].
(2)$$Output={∥x-g\left(f\left(x\right)\right)∥}^{2}$$

### 3.3. Multi-Head Attention Mechanism

Multi-Head Attention is a powerful mechanism that enables the model to concentrate on different parts of the input data at the same time. It is like having several sets of “eyes” focusing on various features or aspects of the data, allowing for the model to capture diverse relationships simultaneously. Recently, the attention mechanism has helped solve many research problems [40]. This mechanism is a core part of the transformer architecture introduced by Vaswani et al., 2017 [41]. Multi-Head Attention is essential because, in simple self-attention, the model learns only one set of attention weights, which can be limiting. For example, a sentence may have a grammatical relationship with one word and a semantic relationship with another. With multiple heads, the model can simultaneously capture these various relationships, leading to better results in natural language processing tasks. In our case, the Multi-Head Attention mechanism is a key component, allowing for the extraction of complex features from sequence data. By utilizing multiple heads, the mechanism can focus on a range of spatial–temporal dependencies within the data, effectively capturing both long- and short-range dependencies. Using our approach, even the slightest patterns in data can be detected, significantly increasing system performance and accuracy.

## 4. The Proposed TT-TBAD Method

Time series anomaly detection is the process of analyzing sets of observations in a multivariable time series $X=\{x_{1},x_{2},\ldots ,x_{N}\}$, with $x_{N}\in R^{O}$ to identify any anomalies. The goal is to develop a scoring function that assigns a higher score to anomalous observations than normal ones $\left(x_{j}\right)$. The input format includes *N* observations, each of *O* variables represented by real numbers. The scoring function Φ assigns a higher score to anomalous observations, which helps detect any unusual behavior $x_{i}$. We have $\Phi \left(x_{i}\right)>\Phi \left(x_{j}\right)$.

### 4.1. Denoising Autoencoder (DAE)

Denoising autoencoders reconstruct the original input from a corrupted version to teach the hidden layer more resilient features. We integrate it into our proposed TT-TBAD to help the model learn the underlying structure of the data and ignore the noise in the training data as shown in Figure 3. The corrupted version is created by adding noise and passed through the encoder to acquire latent representation *h*, as presented by Equations (Equation 3) and (Equation 4). Latent representation *h* is then passed through the decoder to obtain the reconstructed version *x*′ and loss L is computed by taking the difference between the original input x and the reconstructed version *x*′, as presented by Equations (Equation 5) and (Equation 6).
(3)$$\tilde{x}=x+noise$$
(4)$$h=f(\tilde{x})$$
(5)$$x^{\prime }=g\left(f\left(\tilde{x}\right)\right)$$
(6)$$L\left(x,x^{\prime }\right)=\frac{1}{n}\sum_{i=1}^{n}{\left(x_{i}-x_{i}^{\prime }\right)}^{2}$$
where *x* is the original data, $\tilde{x}$ is the data after the noise is added, *x’* is the reconstructed data, and *n* is the dimensionality of the input.

### 4.2. Shared LSTM Encoder with Attention

We designed a shared encoder with LSTM and attention mechanisms. This encoder can process input sequences and generate a context-aware representation that is passed to the model’s numerous Multi-Head Attention mechanisms. For input *X*, the LSTM updates its cell state C and hidden state *h* at each timestep *t*. After processing the sequence, we have a series of hidden states ht. In our case, we are passing these hidden states on to the next phase as presented in Equation (Equation 7).
(7)$$H=[h_{1},h_{2},\ldots ,h_{T}]$$

### 4.3. Attention Mechanism Phase

Attention mechanisms empower models to dynamically allocate significance to different portions of the input sequence, where each sequence is then transformed into *Q*, *K*, and *V* using a “Feature Linear Project”. With the help of these four diverse attention mechanisms, i.e., “Multi-Head Attention”, “Hierarchical Multi-Head Attention”, “Cross-Multi-Head Attention”, and “Feature-Enhanced Multi-Head Attention”, the job is to refine the LSTM−derived hidden states as shown in Equation (Equation 8), each offering a unique perspective on data attention [42]. The detailed representation of model architecture is shown in the figure in Section 4.8.
(8)$$Q=H\times W_{Q},K=H\times W_{K},V=H\times W_{V},$$
where $W_{Q}$, $W_{K}$, and $W_{V}$ are learned weight matrices for queries, keys, and values, respectively. The learned weight matrices by each of these layers have dimensions $d_{model}$ × $d_{model}$, whereas $d_{model}$ is the dimension of each attention. After the transformation, *q*, *k*, and *v* are split into multiple attention heads, as given in Equation (Equation 9). In our case, with $d_{model}$ = 64 and $num_{heads}$ = 8, each head processes a different subspace of the model dimensions, specifically handling $d_{model}$/$num_{heads}$ dimensions, which is 64/8 = 8.

### 4.4. Multi-Head Attention

As previously stated, the transformer architecture employs a mechanism that processes a group of input sequences by projecting them into various subspaces and performing independent attention in each subspace. The Multi-Head Attention algorithm takes in the input sequences in the form of matrices Q(queries), K(keys), and V(values) and operates by conducting the following steps as shown in Figure 4 (left).

Weight matrices multiply input sequences to generate new sequences for each attention head. The attention weights are obtained using a scaled-dot product and passed through Softmax. The final output is the concatenation of all attention head outputs, as presented by Equation (Equation 9).
(9)$$\mathrm{MultiHead}\left(Q,K,V\right)=\mathrm{Attention}\left(QW_{Q},KW_{K},VW_{V}\right)$$

### 4.5. Hierarchical Multi-Head Attention

Different mechanisms are used in the literature to emphasize the special characteristics of the input data. Hierarchical Multi-Head Attention also exhibits the same characteristics. The model employs multiple layers of attention in a hierarchical structure. The output of one layer serves as the query for the next layer, allowing for the model to capture more complex relationships and dependencies between different parts of the sequence, as shown in Figure 5 (right) for hierarchical attention layer *L*, which consists of initial and subsequent layers.

For the first layer, the original *Q*, *K*, and *V* matrices are used. In subsequent layers, the previous layer $output_{l-1}$ is used as the query for the current layer, while *K* and *V* remain the same as shown in Equation (Equation 10).
(10)$$Q_{l}={Output}_{l-1},\;K_{l}=K,\;V_{l}=V$$
where output $L_{1}$ is the output of the hierarchical layer’s $L_{1}$th.

### 4.6. Cross-Multi-Head Attention

The attention mechanism combines standard self-attention with cross-attention, which enhances the queries, keys, and values by projecting additional features. This provides context awareness to the attention mechanism, enabling the model to more effectively learn the dependencies between the different parts of the sequence given in Equation (Equation 11). The original and external keys and values are split into multiple heads, enabling parallel processing and capturing different types of relationships in the data. Moreover, the representation of cross-multi-head attention is visualized in Figure 5 (left). An external source from the LSTM layer provides $V_{\mathrm{external}}$ and $K_{\mathrm{external}}$ shown in the above equation. The final output may consist of a combination of both standard and cross-attention outputs in Equations (Equation 12) and (Equation 13).
(11)$$\mathrm{CrossAttention}(Q,K_{\mathrm{external}},V_{\mathrm{external}})$$
(12)$$\mathrm{Scores}=Q\cdot K_{\mathrm{external}}^{T}$$
(13)$$\mathrm{CrossAttention}\left(Q,K_{\mathrm{external}},V_{\mathrm{external}}\right)=\mathrm{softmax}\left(Q\cdot K_{\mathrm{external}}^{T}\right)\cdot V_{\mathrm{external}}$$

The output is a matrix where each element is a weighted sum of the elements in the external $V_{external}$, with the weights determined by the compatibility of the queries with the keys.

### 4.7. Feature-Enhanced Multi-Head Attention

This mechanism is designed to improve the functionality of queries, keys, and values by adding additional features. These additional features can be in the form of metadata or other relevant data that can be used to provide more context to the data being queried. Doing so makes it easier to retrieve the relevant data and perform more complex operations. Upon instantiation, it creates an additional feature-dense layer designed to project additional features *F*′ into the same dimension as the queries, keys, and values. Feature matrix *F*′ is projected via a weight matrix $W^{F}$ to obtain *F*′, which is then added to *Q*, *K*, and *V* to compute the enhanced attention, as shown in Figure 5 (center). We consider feature matrix *F* in Equation (Equation 14) and the weight matrix for feature projection in Equation (Equation 15).
(14)$$F^{\prime }=FW^{F}$$
(15)$$Q^{\prime }=Q+F^{\prime },\;K^{\prime }=K+F^{\prime },\;V^{\prime }=V+F^{\prime }$$
where $W^{F}$ is a weight matrix for feature projection and the *Q*, *K*, and *V* are the keys that are utilized to compute the attention.

By combining the output of different mechanisms, a comprehensive and nuanced representation of the input data is generated, allowing for more effective analysis and processing in further model layers.

### 4.8. Attention Score Calculation (ASC)

To compute attention scores, each query is compared with all keys. The attention score for each subspace is independently executed using the adaptive feature combination attention approach between query $q_{t}$ and key $k_{t}$, which is calculated for each head and shows the result in Figure 4 (right), determined through specific Equation (Equation 16).

This computation results in a matrix of attention scores, denoted as $s_{t,j}$, which has a size of *T* × *T*, whereas *T* denotes a transpose operation. The value of $s_{t,j}$ represents the score of attending from position *t* to position *j*. $q_{t}$ and $k_{t}$ are query and key vectors at positions *t* and *j* (Equation 17) [43]. A higher score indicates stronger relevance, causing the model to prioritize the corresponding value. Before using the softmax function, the attention score $s_{t,j}$ is substituted by the root of each head ($d_{k}$) to stabilize the gradients, and finally, after passing the result through a Softmax function, the weights are as shown in Equation (Equation 18) [43]:(16)$$\mathrm{ASC}\;s_{t,j}=q_{t}^{T}k_{j}$$
(17)$$Softmax\left(\frac{s_{t,j}}{\sqrt{d_{k}}}\right)$$
(18)$$\mathrm{Attention}\left(Q_{i},K_{i},V_{i}\right)=Softmax\left(\frac{Q_{i}W_{i}^{Q}}{\sqrt{D_{k}}}\right)V_{i}$$
where $D_{k}$ = $d_{model}$/$num_{heads}$ is the depth of each head. When computing attentive representation and concatenation for each timestep *t*, the weighted sum of all values is presented in Equation (Equation 19). In this way, we obtain a sequence of attentive representations $R=[r_{1},r_{2},r_{3},r_{4},\ldots ,r_{n}]$, as presented in Equation (Equation 20). The above process is repeated N times (for Nheads) with different weight matrices $W_{Q}^{n}$, $W_{K}^{n}$, $W_{V}^{n}$ each time and produces attentive representation sequence $R_{n}$ differently as a final output, as given in Equation (Equation 21). Here, we acquire the output of all attention mechanisms after concatenation for the concatenated output, as presented in Equation (Equation 22) which can be seen in the detailed model architecture; Figure 6.
(19)$$r_{t}=\sum_{j=1}^{T}a_{t,j}v_{j}$$
(20)$$Concat=[r_{1},r_{2},r_{3},r_{4},\ldots ,r_{n}]$$
(21)$$O_{t}=\left[R_{t}^{1},R_{t}^{2},R_{t}^{3},\ldots ,R_{t}^{N}\right]\times W_{O}$$
(22)ConcatOutput=Concat(MHA,H-MHA,C-MHA,FE-MHA)
where in Equation (Equation 21) $W_{O}$ is another learned weight matrix, and Equation (Equation 22) denotes the combination of outputs from multiple attention mechanisms into a single tensor. In this case, the second shared encoder evaluates the merged output, capturing intricate temporal patterns, assimilating information from various attention outputs, and transmitting it to the twin branch.

### 4.9. Reconstruction and Prediction Based on Encoded Representation with Attention

The input sequence *X* is employed in condensed representation *E* by encoding function $f_{enc}$, which likely employs attention mechanisms to capture significant features of *X*.

Two mappings denote the twin branch (reconstruction branch, prediction branch); as shown in the architecture of the proposed model Figure 6, the first one reconstructs *X* from *E* and is represented as $X=f_{Rec}\left(E\right)$. The second mapping predicts future *X* values from *E*, denoted as $P=f_{pred}\left(E\right)$.

#### 4.9.1. Reconstruction Branch

To measure the error in the reconstruction process, we use reconstruction error $L_{rec}$, which calculates the discrepancy between *X* and its reconstructed version $\hat{X}$, across all timesteps using the Euclidean $L_{2}$ norm. This essentially computes the geometric distance between the corresponding elements of *X* and $\hat{X}$ at each timestep *t* and adds up these distances as shown in Equation (Equation 23). The LSTM encoder takes a noisy input sequence, $X_{noisy}$ with *t* timesteps and *F* features and produces a hidden representation, $H_{E}$, in Equation (Equation 24). This encoded sequence $H_{E}$ is then repeated to match the original sequence length *L* as follows in (Equation 25). The repeated encoded sequence $H_{\mathrm{rp}}$ is applied on the layer (decoder) to generate a new hidden representation $H_{D}$ as given in Equation (Equation 26). Finally, hidden representation $H_{D}$ is processed with a TimeDistributed Dense layer to reconstruct sequence $X_{\mathrm{Rec}}$ as follows in Equation (Equation 27).
(23)$$L_{\mathrm{Rec}}=\sum_{i=1}^{t}\left|\right|X_{i}-{\hat{X}}_{i}{\left|\right|}^{2}$$
(24)$$H_{E}=\mathrm{LSTM}\left(X_{\mathrm{noisy}}\right)$$
(25)$$H_{\mathrm{rp}}=\mathrm{Repeat}\left(H_{E},L\right)$$
(26)$$H_{D}=\mathrm{LSTM}\left(H_{\mathrm{rp}}\right)$$
(27)$$X_{\mathrm{Rec}}=\mathrm{TimeDistributed}(\mathrm{Dense})\left(H_{D}\right)$$

#### 4.9.2. Prediction Branch

Similarly, to measure the error in the prediction process, we use prediction error $L_{pred}$ which measures the discrepancy between predicted future values *P* and actual future values $X_{future}$ over a defined future timeframe as shown in (Equation 28). The geometric distance is again used to quantify the discrepancy at each future timestep $T_{f}$, and these distances are summed up. The representation by encoder $H_{E}$ undergoes processing via Dense layers to predict future values. The intermediate representation is $H_{\mathrm{predDense}}$, and the final predicted values are $Y_{\mathrm{pred}}$ as given in Equations (Equation 29) and (Equation 30).
(28)$$L_{\mathrm{pred}}=\sum_{i=1}^{\mathrm{future}\_\mathrm{timesteps}}\left|\right|X_{\mathrm{future},i}-P_{i}{\left|\right|}^{2}$$
(29)$$H_{\mathrm{predDense}}={\mathrm{Dense}}_{\mathrm{relu}}\left(H_{E}\right)$$
(30)$$Y_{\mathrm{pred}}=\mathrm{Dense}\left(H_{\mathrm{predDense}}\right)$$

## 5. Dataset Analysis and Explanation

The datasets as shown in Figure 7 and Figure 8 are composed of multiple time series produced by different sensors. These sensors help us maintain a comfortable environment for the pigs. We installed several sensors inside our pig barn to monitor temperature (°C), humidity (%), carbon dioxide CO_2_ (ppm), ammonia NH_3_ (ppm), and hydrogen sulfide H_2_S (ppm).

We installed weather forecasting devices outside to obtain accurate weather readings. These devices include temperature (°C), humidity (%), carbon dioxide CO_2_ (ppm), ammonia NH_3_ (ppm), and hydrogen sulfide H_2_S (ppm). By monitoring both the inside and outside environment, we can make informed decisions to ensure the well-being of our pigs, as shown in Figure 9. Our setup consists of interconnected sensors that generate copious amounts of data, up to 39,456 data points per sensor. Additional details are found in Table 1.

All these sensors converge at a local server, Jetson Nano, where the data are immediately stored as they are collected. Acting as an intermediate repository and processing node, this Jetson Nano is then connected to a central server, which serves as a comprehensive data hub for all the collected data. This includes sensor data, imagery, and video feeds from cameras, all securely stored in real-time to ensure seamless consolidation, correlation, and analysis. As mentioned in the introduction in Section 1, the dataset is sourced from our pig barn farm project. The sensors produce data with a 5-min delay, creating a multivariate time series of nine features. These features include information about the indoor conditions of the barn, such as fattening and outdoor weather forecasts. For more detailed information, please refer to Table 1.

### 5.1. Data Analysis and Correlation

In livestock farming, environmental parameters play a vital role in animal health, growth, and productivity. Even minor changes in temperature or CO_2_ levels can drastically affect the well-being of these animals as shown in Figure 10. Therefore, it is crucial to study moderate correlations that may seem insignificant at first glance.

### 5.2. Outdoor Weather Station Correlation

Our research shows that temperature and humidity are negatively correlated, indicating that when temperature increases, humidity tends to decrease, and vice versa. On the other hand, solar radiation is positively correlated with cumulative solar radiation, which is expected as the latter is a cumulative measure. The other features do not exhibit strong correlations as shown in Figure 10 (left).

### 5.3. Fattening House Indoor Correlation

Figure 10 (right) shows that temperature and humidity are negatively correlated, which is the same as at the outdoor weather station, with similar data for outdoors. CO_2_ concentration has a slight negative correlation with humidity. The other features in the fattening house data do not have strong correlations.

When analyzing data, it is crucial to consider the correlation between variables. The strength of the correlation can be classified into three categories.

#### 5.3.1. Weak Correlation

Even if the correlation is weak, it should still be considered if the variable has practical significance, such as CO_2_ levels. General interpretations for the strength of the correlation coefficient (*rr*) can be
(31)$$\left(|rr|<0.5\right)$$

#### 5.3.2. Moderate Correlation

Moderate correlations should be closely examined, as their combined effects or interactions with other variables may be important as shown in Equation (Equation 32).
(32)0.5≤∣rr∣<0.7

#### 5.3.3. Strong Correlation

Strong correlations are obviously significant and should be included in the analysis. However, if using linear models, caution should be observed regarding multicollinearity as given in Equation (Equation 33).
(33)(|rr|≥0.7)

## 6. Experimental Results and Discussion

### 6.1. Implementation Details

We developed and tested anomaly detection models for multivariate time series using a Shared LSTM Twin-Branch technique and a Multi-Head Attention mechanism. The models were implemented in Python 3.7 using popular frameworks such as Pandas, NumPy, Scikit-Learn, TensorFlow 2, and the Keras API. The experiments were performed on a PC equipped with an Intel (R) CoreTM i7-10700 CPU @ 2.90 GHz and 32 GB of RAM. The model architecture parameters are explained in Table 2.

### 6.2. Performance Metrics

The process of evaluating an unlabeled anomaly detection model involves using various performance metrics, including mean squared error (MSE), mean absolute error (MAE), root mean squared error (RMSE), and the $R^{2}$ score as shown in Equations (Equation 34)–(Equation 37). These metrics are instrumental in the accurate assessment of the model’s performance. While MSE and MAE capture the reconstruction errors vital for anomaly detection, RMSE provides additional information on the magnitude of errors. In addition, the $R^{2}$ score, although unusual in anomaly detection, helps in comparing the ability of our model to capture data variance with that of baseline models as given in the equations below. This comprehensive set of metrics ensures that our model’s anomaly detection performance is robust, even in the absence of ground truth labels [11].
(34)$$\mathrm{MSE}=\frac{1}{n}\sum_{i=1}^{n}{\left(y_{i}-{\hat{y}}_{i}\right)}^{2}$$
(35)$$\mathrm{MAE}=\frac{1}{n}\sum_{i=1}^{n}\left|y_{i}-{\hat{y}}_{i}\right|$$
(36)$$\mathrm{RMSE}=\sqrt{\frac{1}{n}\sum_{i=1}^{n}{\left(y_{i}-{\hat{y}}_{i}\right)}^{2}}$$
(37)$$R^{2}=1-\frac{\sum_{i=1}^{n}{\left(y_{i}-{\hat{y}}_{i}\right)}^{2}}{\sum_{i=1}^{n}{\left(y_{i}-\bar{y}\right)}^{2}}$$

The model parameters and their respective values are detailed in Table 2. We split the raw data into training and validation sets to train our model, with an 80% and 10% split, respectively. Since we do not have separate data for testing, we set aside 10% of the data for testing the model, as given in Table 1. To normalize the data, we utilized the MinMaxScaler from the scikit–learn library in Python, which scales each feature to a specified range, usually between zero and one. Normalization guarantees that each feature contributes equally to the final prediction, which is particularly useful when dealing with features of varying magnitudes. It also aids in model convergence during training and ensures accurate model performance evaluation through reported metrics such as MSE, MAE, RMSE, and $R^{2}$ Score across all features. The batch sizes of the dataset used for training the model varied from 24 to 32. To enhance the ability of the model to learn data representation and make the training more robust, we included Gaussian noise using DAE in the training set. The progress of model training was analyzed using a sophisticated monitoring technique known as the feedback loop paradigm [44], which allowed for a meticulous examination of the process. The reconstruction and prediction errors were comprehensively evaluated throughout each epoch and systematically recorded. These errors were visualized as a bar graph, revealing feature-specific and iteration-specific nuances, as demonstrated in Figure 11. This in-depth analysis facilitated a nuanced understanding of the model’s learning dynamics, providing insights into the complex challenges encountered and overcome during training. Eventually, the model was successfully trained on the validation set after employing specific techniques.

During the training of our model, we evaluated its performance on the validation dataset for each epoch. The graph in Figure 12 shows the model performance on the indoor fatten barn dataset and the outdoor weather forecast dataset. The graph highlights that the model learned quickly at the beginning of the training process, but the loss curves stabilized in the later epochs to avoid overfitting. We employed the *kernelregularization* technique to prevent overfitting, which is elaborated in Table 2. Additionally, we applied the early stopping technique to save computational resources.

### 6.3. Anomaly Detection and Prediction for the Indoor Fattening Barn Environment Dataset

Our model is trained using a shared LSTM encoder with several Multi-Head Attention mechanisms and reconstructed with Twin-Branch techniques, which utilize reconstruction errors to identify abnormalities and prediction errors that enable accurate predictions of the following values in the sequence during runtime. The method assigns loss weights of 0.5 and 1.0, respectively, for the two branches involved in the process.

We comprehensively evaluated our proposed method by comparing it with several baseline models. To ensure a fair comparison between different models, we made sure to maintain identical experimental conditions. This involved using the same data splits for training and testing, applying the same preprocessing routines, and utilizing the exact architectures and parameters as specified in the original papers. This ensured that any observed differences in performance were solely due to the inherent properties of the models themselves, rather than any discrepancies in the methodology used to evaluate them. This validates the credibility of our approach and facilitates a fair comparison, underscoring the reliability and thoroughness of our study. TT-TBAD was conducted and compared against various state-of-the-art baseline models across multiple features. Table 3 displays the MSE, MAE, RMSE, and $R^{2}$ scores for each feature. Our proposed “TT-TBAD” methodology outperforms the other analyzed baseline models. Regarding predicting humidity, our model achieves an over $R^{2}$ score = 0.93 accuracy in almost all feature cases except for H_2_S.

This could indicate potential data discrepancies. Unlike other models such as Attention and Bi-LSTM [39], which struggle with the complexities of their attention mechanisms, our model utilizes the power of the Multi-Head Attention technique and achieves higher precision. Through this distinction, the strength and ingenuity of our approach are highlighted.

After conducting a thorough collective comparison with other baseline models, as shown in Table 4 and Table 5, our advanced TT-TBAD technique achieved a remarkable performance $R^{2}$ score of 0.993 and 0.994, indicating its superior predictive capabilities. On the other hand, the “Attention Bi-LSTM” model falls behind. In addition, traditional approaches like LSTM [22] are not as effective, as their high error rates illustrate the challenges they face in understanding the complex dynamics of the barn environment.

We used the dynamic threshold approach based on rolling mean addition to *k* standard deviation to determine the threshold for the reconstruction error. This is a flexible method for anomaly detection in time series data. It can adjust to changing data patterns and is suitable for many applications, especially in dynamic environments, as shown in Equation (Equation 38). This approach is more effective than the traditional technique of taking the average of the five highest loss values from the validation set [45]. Our model can identify anomalies in the data by using the predetermined dynamic threshold. If the reconstruction loss exceeds this threshold, the model will recognize those values as abnormal; Ref. [39], as shown in Figure 13.
(38)$$\mathrm{Dynamic}\;{\mathrm{Threshold}}_{t}={\mathrm{RollingMean}}_{t}+k\times {\mathrm{RollingStdDev}}_{t}$$

Meanwhile, the RollingMean represents the reconstruction error, and the RollingStdDev is the standard deviation of observations over a specified window and time. *k*, a predefined constant, determines the number of standard deviations from the rolling mean required to qualify a point as an anomaly.

Smoothing the errors by averaging during the rolling mean calculation over a specified data window helps to reduce the reconstruction error. As a result, the green line representing the rolling mean appears smooth. Validation data were utilized for testing purposes, and we present the results in Figure 14.

The figure includes five line charts that display the comparison between the anomalies and actual values for each feature separately, based on a small portion of the validation set. Each chart plots data points over a common x-axis, with several samples for each feature, including temperature, humidity, CO_2_, NH_3_, and H_2_S. After analyzing 39,456 samples, we identified anomalies in varying percentages across different metrics. The temperature metric showed a 3.29% anomaly rate, humidity at 3.39%, CO_2_ at 3.58%, and NH_3_ stood out with a higher rate of 4.19%.

Interestingly, H_2_S displayed the lowest anomaly rate at 2.62%. This is due to the model’s higher reconstruction and prediction error, with an $R^{2}$ score = 0.71. In comparison, other features exhibited metrics indicating better performance. This discrepancy emphasizes how model accuracy and prediction errors can impact anomaly detection.

The model predictions for temperature and humidity in the prediction phase are quite accurate, indicating its reliability. However, there are some noticeable discrepancies in temperature around indices 200, 600, and 800 and humidity near 800, which may be due to external factors or model limitations. CO_2_ predictions match well with actual data, but some deviations at indices 200 and 600 could be attributed to environmental changes. On the other hand, predicting NH_3_ and H_2_S levels is more challenging, with more pronounced fluctuations, particularly around indices 200–800 for NH_3_ and 200–400 for H_2_S, highlighting potential challenges or sporadic environmental events. Moreover, predicting the lines using H_2_S presents challenges, making the model’s future job more challenging. The deviations in these metrics underscore the areas for model improvement, as depicted in Figure 15.

### 6.4. Comparative Analysis of Anomaly Detection in Indoor Environment

The proposed TT-TBAD for anomaly detection is evaluated against state-of-the-art models, including LSTM-AE, GRU, LSTM, ConvLSTM-AE, and Attention-Bi-LSTM, with an assessment based on accuracy [46], precision [47], recall [48], and the F1 score [49]. The comparative study involves the utilization of a data annotation mechanism to label the partial test dataset, ensuring suitability for the detection evaluation. The results are presented in Figure 16, revealing that the LSTM-AE model demonstrated a robust balance between precision (0.867) and recall (0.853), resulting in a solid F1 score of 0.868. The GRU slightly outperformed, achieving precision and recall values of 0.87 and 0.871, respectively, although there was a modest dip in the F1 score to 0.844. The standard LSTM lagged behind, with precision and recall scores of 0.824 and 0.835, hinting at a potential decrease in anomaly classification accuracy, as reflected by an F1 score of 0.809. In contrast, the ConvLSTM-AE model excelled, boasting precision and recall scores of 0.881 and 0.879, along with an impressive F1 score of 0.893, indicating enhanced anomaly detection capability. Additionally, the Attention-Bi-LSTM model achieved the highest precision at 0.91 and a recall of 0.902, resulting in an F1 score of 0.891, showcasing strong predictive confidence. However, the spotlight is on our TT-TBAD model, surpassing all others with outstanding precision (0.94), recall (0.913), and F1 score (0.92), coupled with an unmatched accuracy of 0.983. This establishes a new standard in anomaly detection, highlighting our model’s exceptional precision and reliability.

The Receiver Operating Characteristic (ROC) curve highlights the effectiveness of environmental classifiers in detecting gases within an enclosed setting. Among the classifiers, temperature demonstrated the highest performance with an AUC score of 0.98. NH_3_, CO_2_, and H_2_S classifiers also performed exceedingly well with AUC scores ranging from 0.96 to 0.97. The humidity classifier had an AUC score of 0.96, indicating its potential effectiveness in detecting gases.

These AUC values presented in Figure 17, markedly higher than the random guess baseline (AUC = 0.5), reflect the classifiers’ high accuracy and confirm their potential for reliable indoor air quality monitoring, offering substantial benefits for safety and hazard prevention in controlled environments.

### 6.5. Anomaly Detection and Prediction for Outdoor Weather Forecast for the Barn Environment Dataset

Utilizing the same effective methodology employed in the previous dataset, we applied our method with great attention to detail to ensure uniformity and comparability in our analysis. Upon thoroughly examining Table 6, it is evident that the TT-TBAD method consistently outperforms its counterparts.

The performance of various models was evaluated based on different features such as temperature, humidity, solar energy, and accumulated solar energy per day. The TT-TBAD method displayed superior precision for temperature and humidity, with the lowest MSE, MAE, and RMSE values and $R^{2}$ scores = 0.99 and 0.99, respectively. In solar energy, the TT-TBAD method outperforms other models with its impressive $R^{2}$ score = 0.94 for Attention-Bi-LSTM and 0.98 for accumulated solar energy/day. Although Attention-Bi-LSTM has slightly better error metrics, TT-TBAD stands out for its adaptability and resilience, making it a reliable and versatile model for predictive analysis. Its consistency in performance across various datasets underlines its potential as an excellent choice for predictive analysis of solar energy.

The study used various modeling techniques for outdoor weather forecasting in a fattening barn environment. The results showed that our TT-TBAD method performed the best with the lowest mean squared error (MSE) of 0.000618 and an excellent $R^{2}$ Score = 0.99. This demonstrates the exceptional predictive accuracy of our method in this particular setting. Other methods, such as LSTM-AE and ConvLSTM-AE, also performed well, with $R^{2}$ Scores = 0.95 and 0.96, respectively. However, Attention-Bi-LSTM achieved the highest $R^{2}$ score = 0.97. GRU slightly surpassed LSTM-AE with an $R^{2}$ score = 0.90.

Figure 18 illustrates the same dynamic threshold approach is used for outdoor weather forecast datasets to identify anomalies in the data compared to the reconstruction error. The black data points indicate the anomalies, meaning the model finds it difficult to reconstruct them accurately. The blue line represents the difference between the original and reconstructed data.

The solar energy graph has pronounced fluctuations with several anomalies, and the accumulated solar energy chart has a step-like pattern, with anomalies signifying unexpected drops or surges. These graphs emphasize the need to monitor environmental and energy parameters for consistency. The graphs in Figure 19 compare actual and predicted values. While the model predictions for temperature and humidity are highly accurate, noticeable discrepancies are evident in the solar energy graph. These inconsistencies underscore the imperative need to develop and refine prediction mechanisms, particularly in the context of solar energy parameters. It is crucial to address these discrepancies to enhance the accuracy of the model predictions.

Upon analyzing the data using the rolling mean approach, it becomes evident that metrics related to solar energy, such as solar energy and accumulated solar energy per day, exhibit a slightly higher percentage of anomalies, approximately 4.43% and 4.54%, respectively, as shown in Figure 20. In contrast, anomalies in temperature and humidity are only slightly lower, at around 3.36% and 3.44%, respectively. This discrepancy suggests that the solar metrics may be more susceptible to abnormalities or external factors influencing their readings when compared to the more consistent readings of temperature and humidity.

### 6.6. Comparative Analysis of Anomaly Detection Models in Outdoor Environment

A comparative study of anomaly detection is performed for various models, and the results are presented in Figure 21. The figure shows that the LSTM-AE model achieves a solid precision rate of 0.842 paired with a recall of 0.83, harmonizing into an F1 score of 0.848. Nipping at its heels, the GRU model registers a precision of 0.83 with an incrementally higher recall of 0.851, achieving an F1 score of 0.841. Transitioning to the LSTM model, we observe a dip in precision to 0.792 and recall to 0.809, reflected in a modest F1 score of 0.81. In contrast, the ConvLSTM-AE model emerges stronger, with a precision of 0.862 and recall of 0.859, coalescing into a solid F1 score of 0.876. A leap forward is seen with the Attention-Bi-LSTM model, which achieves a notable precision of 0.909 and a recall of 0.899, commanding an F1 score of 0.901. The crescendo of our analysis is reached with the introduction of our TT-TBAD model. It sets a new benchmark with a precision of 0.927 and a recall of 0.909, melding into an F1 score of 0.91. This performance is further underscored by a remarkable accuracy of 0.971, positioning our model at the apex of anomaly detection capabilities.

Moreover, the ROC curve presented in Figure 22 demonstrates the discriminative power of four classifiers based on environmental parameters. The figure shows AUC scores of 0.97 for both temperature and humidity, followed closely by accumulated solar energy per day at 0.96 and solar energy at 0.94. The classifiers exhibit robust predictive capabilities, with their performance significantly surpassing the random classification benchmark (AUC = 0.5). This underscores their potential utility in sophisticated environmental sensing and prediction applications.

## 7. Discussion

The TimeTector-Twin-Branch Shared LSTM encoder (TT-TBAD) is a revolutionary development in the field of agricultural data analytics. It is particularly useful for detecting anomalies in time series data, which can be noisy and difficult to analyze. This model is designed to understand agricultural datasets and redefine the standards of anomaly detection by identifying genuine irregularities that are crucial for operational farm decision making.

At its core, the proposed method uses a shared LSTM encoder, which is known for its ability to comprehend temporal data, along with advanced attention mechanisms. These mechanisms filter through data to eliminate irrelevant noise and highlight critical anomalies. This is especially important in agriculture, where distinguishing between benign fluctuations and significant outliers is essential for crop yield and livestock health. TT-TBAD also uses a Denoising Autoencoder (DAE) to counter the challenge of overfitting. This ensures that the model remains robust and generalizable, even when faced with new data. The Twin-Branch design of the model allows for it to simultaneously reconstruct data and predict future environmental and biological shifts, providing valuable predictive foresight for farmers which traditional models failed to provide. What sets TT-TBAD apart is its ability to balance architectural complexity with computational efficiency. Despite its advanced design, TT-TBAD has commendable training durations between 380 and 400 s, meaning that it respects the time-sensitive demands of agricultural operations. This efficiency is achieved without sacrificing the depth and breadth of its data analysis, maintaining high fidelity in anomaly detection and prediction. TT-TBAD introduces a dynamic thresholding technique based on a rolling mean, which adapts to evolving data patterns. This fine-tunes the model’s sensitivity to fluctuations and enhances its accuracy, making it an indispensable asset in precision farming technologies. In sum, TT-TBAD is an excellent example of meticulous data analysis, advanced predictive modeling, and practical computational application. Its unmatched accuracy in detecting and predicting anomalies, underpinned by a dynamic and adaptive thresholding strategy, makes it a valuable tool for precision farming technologies that work in harmony with nature and machine learning innovation.

### Limitations and Assumptions

The model is tested and found to be efficient in identifying any unusual or abnormal data points within the current datasets. However, its scalability and complexity make it challenging to process larger datasets, which requires longer training periods and more computational power. Moreover, parameter tuning has become more complex, and refined optimization techniques are necessary. The model’s anomaly detection capability depends on the quality of the input data, and it is crucial to judiciously integrate noise to preserve its effectiveness. The model’s design is based on fundamental assumptions about input data and training parameters, such as noise levels, sequence sizing, and batch processing, which aim to improve its performance in detecting anomalies within agricultural contexts.

## 8. Conclusions

In our research paper, we introduce the TimeTector-Twin-Branch Shared LSTM encoder (TT-TBAD) model which employs a Twin-Branch Shared LSTM encoder and multiple Multi-Head Attention mechanisms to detect anomalies and make real-time predictions from multivariate time series sensor data. With the advent of deep learning algorithms, the challenge of capturing spatial–temporal correlations in multi-sensor time series data is efficiently addressed. These algorithms excel at generalizing normal, noisy, and abnormal data patterns, making them an ideal solution for handling complex data structures. Our model is a fusion of several robust frameworks. It incorporates the Denoising Autoencoder (DAE) to characterize multi-sensor time series signals, thus minimizing the risk of overfitting caused by noise and anomalies in the training data. Additionally, we utilize various Multi-Head Attention mechanisms such as Multi-Head Attention (MHA), Cross-Multi-Head Attention (C-MHA), Hierarchical Multi-Head Attention (Hi-MHA), and Feature-Enhanced Multi-Head Attention (FH-MHA). These mechanisms enable the model to capture diverse and hierarchical relationships, recognize cross-sequence patterns, and leverage external features, resulting in a comprehensive understanding of complex datasets. Our experimental evaluation is conducted on proprietary livestock barn environmental datasets using performance metrics for both anomaly detection and future prediction, where our model demonstrates outstanding performance compared to benchmark traditional models with detection accuracies and $R_{2}$ scores of 97.1%, 98.3% and 0.994%, 0.993%.

In our future work, we aim to develop a smart agriculture livestock environment that allows real-time monitoring through the barn sensors. The ultimate aim is to automate livestock farming to the greatest possible extent.

## Figures and Tables

**Figure 1 sensors-24-02453-f001:**
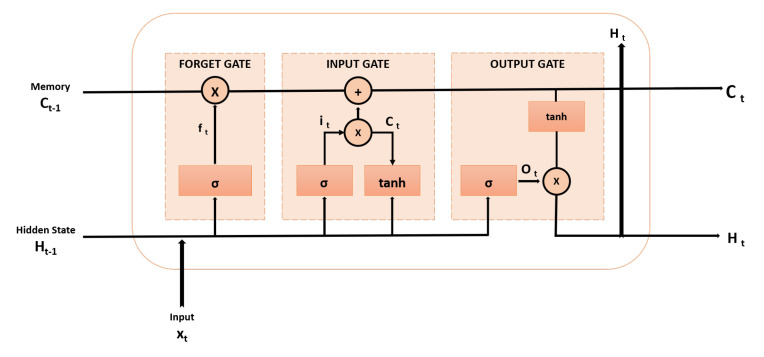
Long short-term memory (LSTM) cell with an input gate, an output gate, and a forget gate [39].

**Figure 2 sensors-24-02453-f002:**
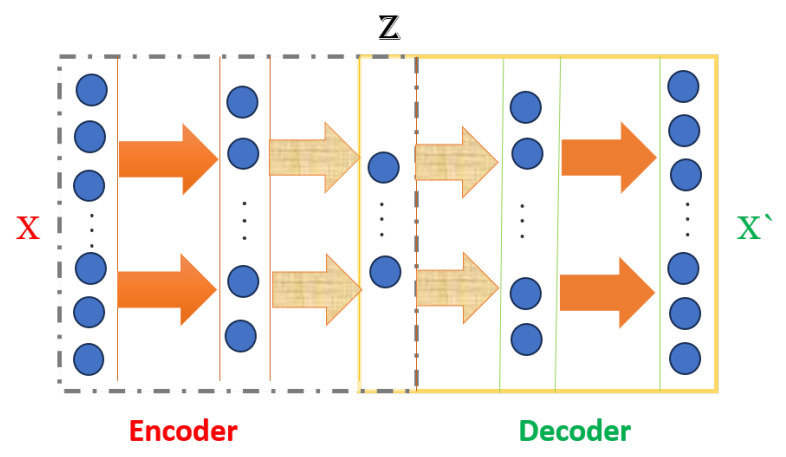
Image illustrates an encoder –decoder neural network model. The encoder compresses the input ‘X’ into a condensed representation ’Z’, which the decoder then uses to reconstruct the output ‘X prime.’ Arrows indicate process flow and blue dots represent distinct layers [21].

**Figure 3 sensors-24-02453-f003:**
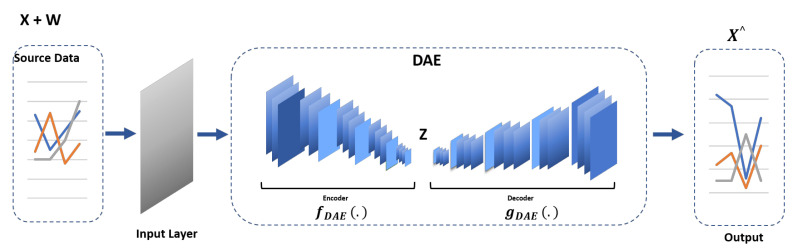
The denoising autoencoder architecture shows the addition of noise to the data passing through the encoder–decoder and subsequently through the shared LSTM.

**Figure 4 sensors-24-02453-f004:**
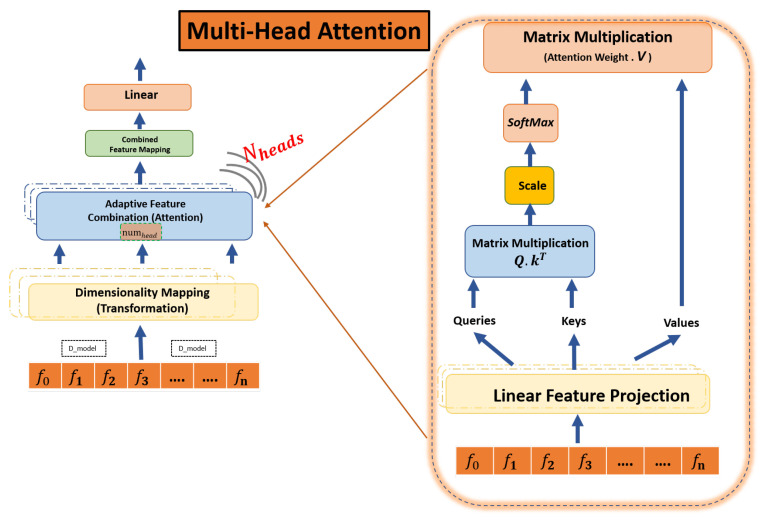
Structure of the Multi-Head Attention module: (**left**) the multi-head adaptive feature combination attention module with *N* parallel heads; (**right**) the basic scaled-dot product attention in each head separately.

**Figure 5 sensors-24-02453-f005:**
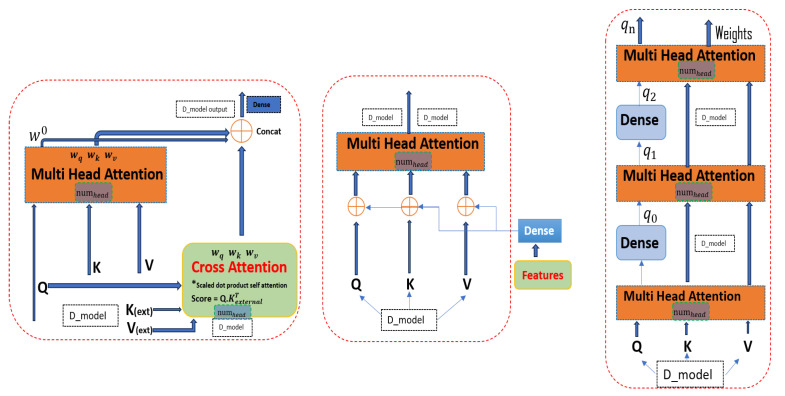
Structure of the Cross-Multi-Head Attention module (**left**), Feature-Enhanced Multi-Head Attention module (**center**), and Hierarchical Multi-Head Attention module (**right**).

**Figure 6 sensors-24-02453-f006:**
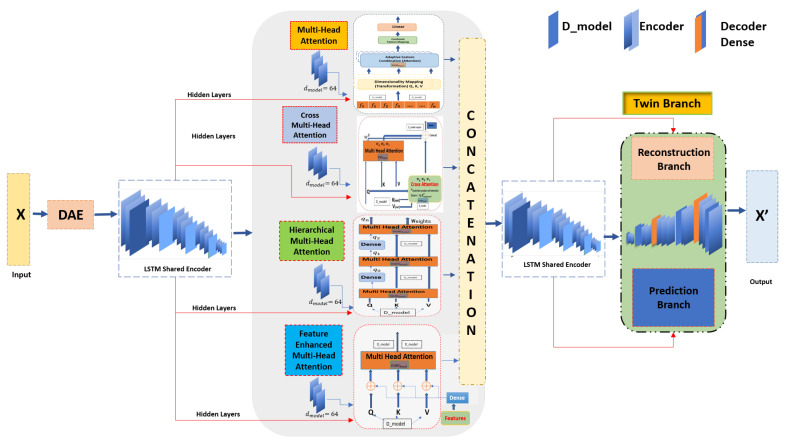
Architecture of the proposed TT-TBAD highlighting two shared LSTM encoders and an attention branch portion leading to the Twin-Branch section.

**Figure 7 sensors-24-02453-f007:**
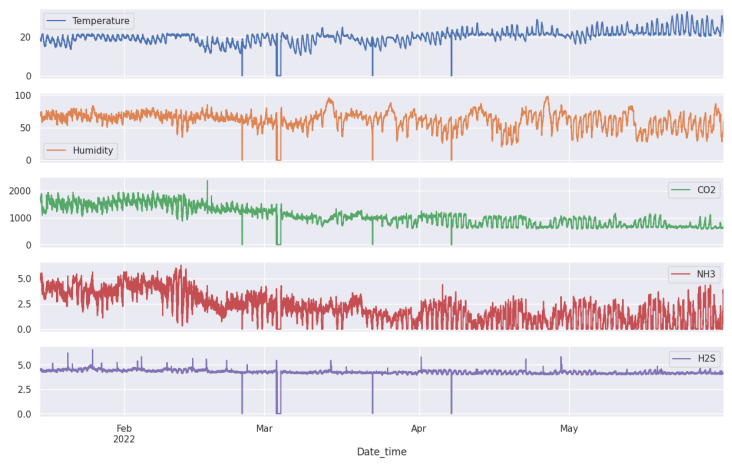
Fattening data collected indoors from a pig farm environment includes measurements of temperature, humidity, CO_2_, NH_3_, and H_2_S.

**Figure 8 sensors-24-02453-f008:**
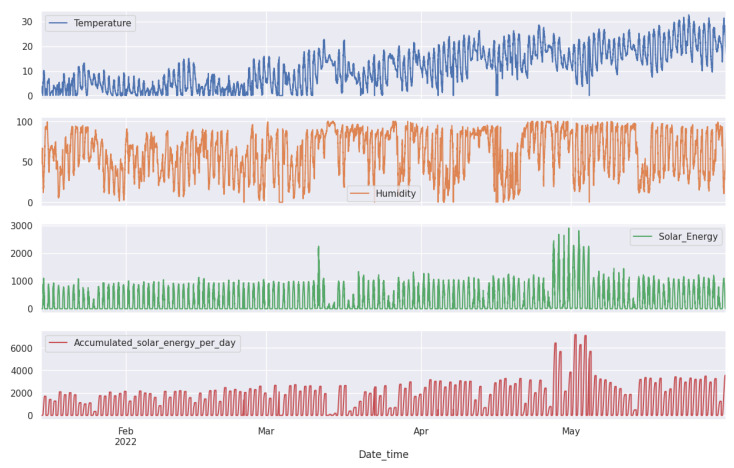
Outdoor data retrieved from the weather forecast station at the pig farm environment includes temperature, humidity, solar energy, and accumulated solar energy per day.

**Figure 9 sensors-24-02453-f009:**
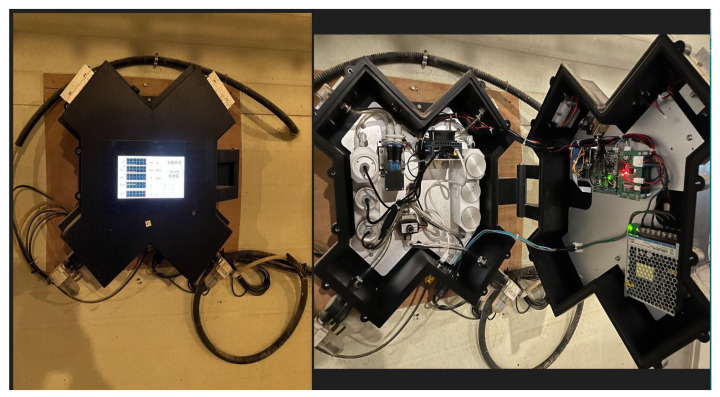
Experimental setup involves the use of sensor detectors placed in a box equipped with a Jetson Nano board for monitoring indoor and outdoor environmental conditions in a pig farm.

**Figure 10 sensors-24-02453-f010:**
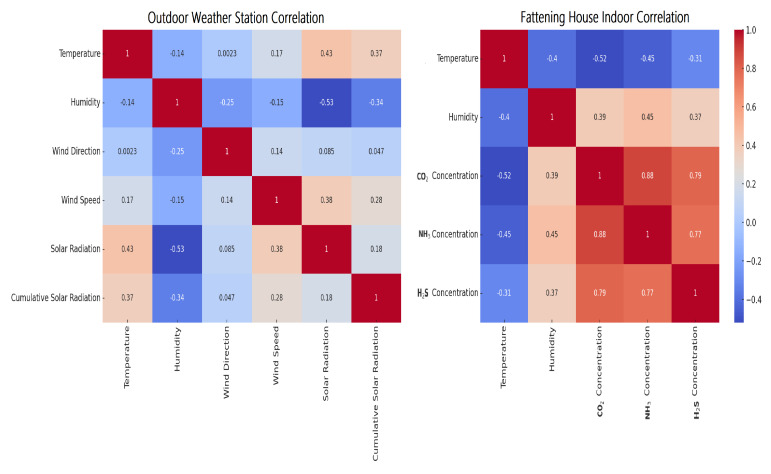
Confusion matrices for outdoor (**left**) and indoor (**right**) pig farm datasets.

**Figure 11 sensors-24-02453-f011:**
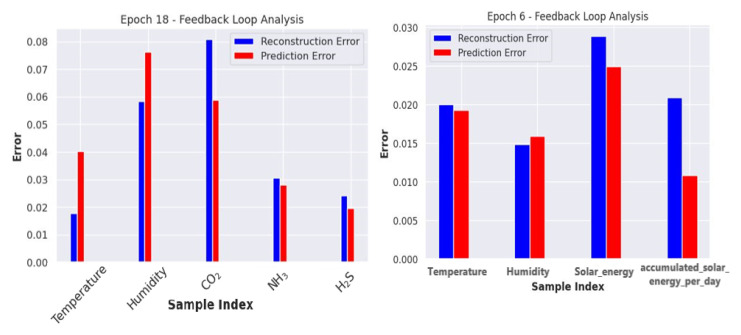
Feedback loop at each iteration for the indoor (**left**) fattening dataset and outdoor (**right**) weather forecast dataset.

**Figure 12 sensors-24-02453-f012:**
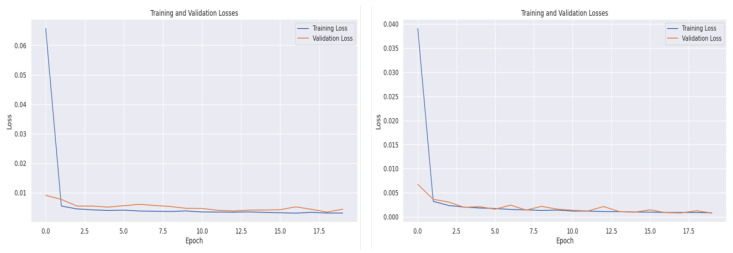
A representation of the training and validation loss for the indoor (**left**) fattening barn dataset and outdoor (**right**) weather forecast dataset.

**Figure 13 sensors-24-02453-f013:**
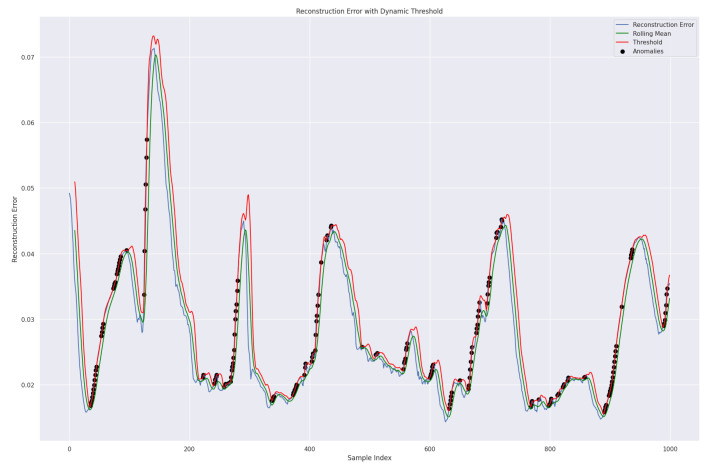
A representation of the proposed TT-TBAD model for anomaly detection with a dynamic threshold based on the rolling mean and reconstruction error over a specified window and time for the indoor fattening barn environment.

**Figure 14 sensors-24-02453-f014:**
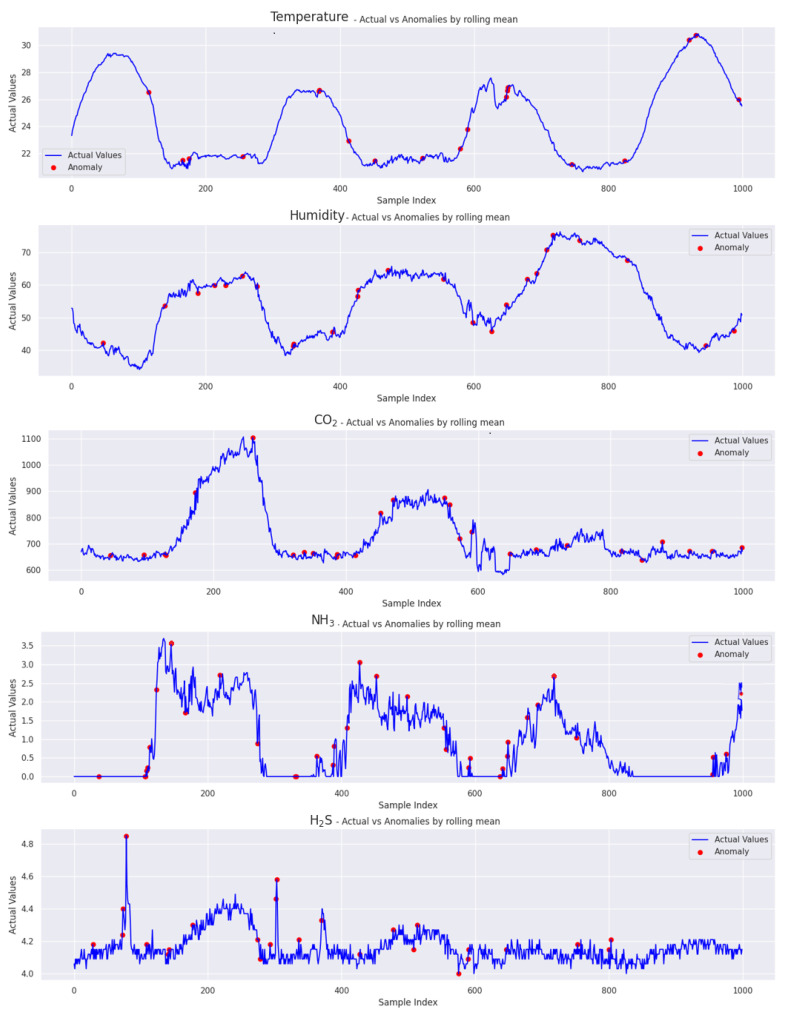
A representation of the proposed TT-TBAD model for anomaly detection with a dynamic threshold based on the rolling mean and reconstruction error over a specified window and time for each feature in the indoor fattening barn environment.

**Figure 15 sensors-24-02453-f015:**
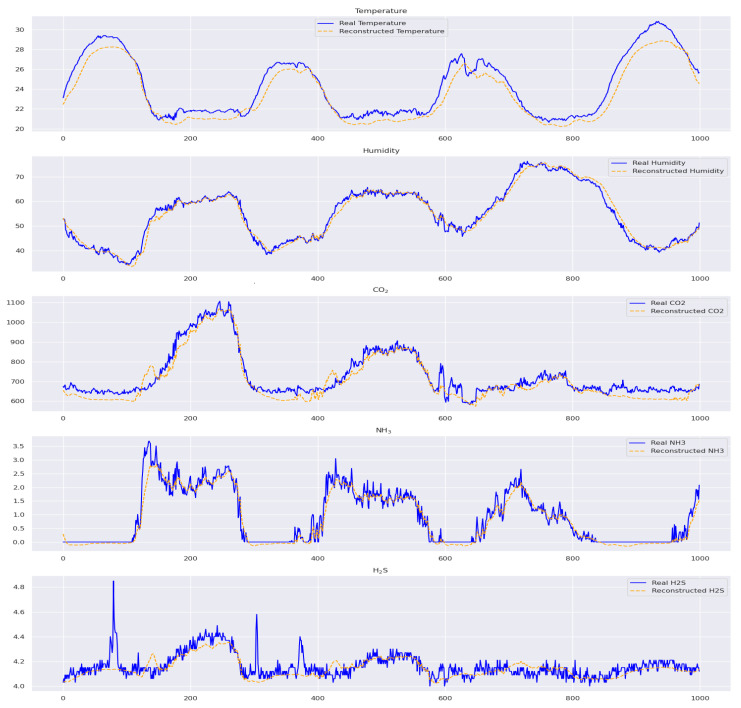
A comparison of the proposed TT-TBAD model for actual and predicted data points in the indoor fattening barn environment.

**Figure 16 sensors-24-02453-f016:**
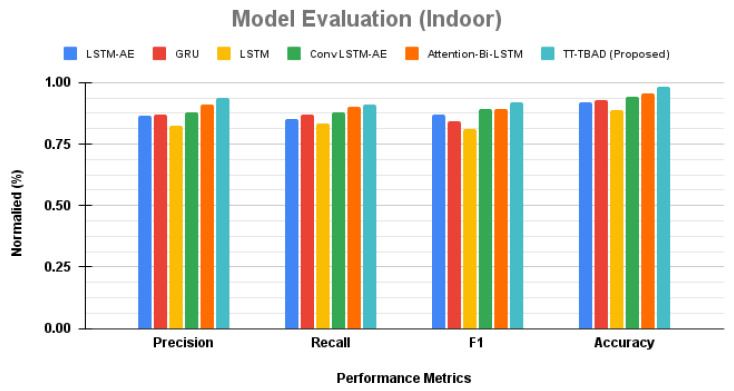
A comparative study of the proposed TT-TBAD for anomaly detection in indoor environments against various state-of-the-art anomaly detection models.

**Figure 17 sensors-24-02453-f017:**
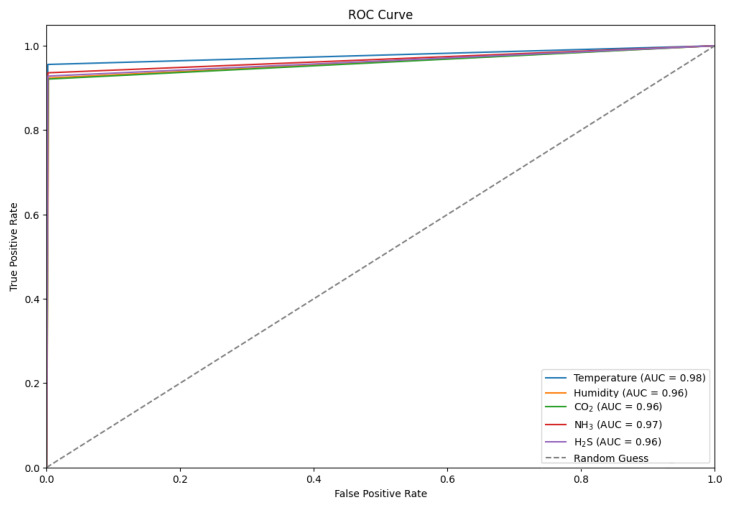
A ROC curve of the proposed TT-TBAD for five classifiers based on environmental parameters of the barn dataset in an indoor environment.

**Figure 18 sensors-24-02453-f018:**
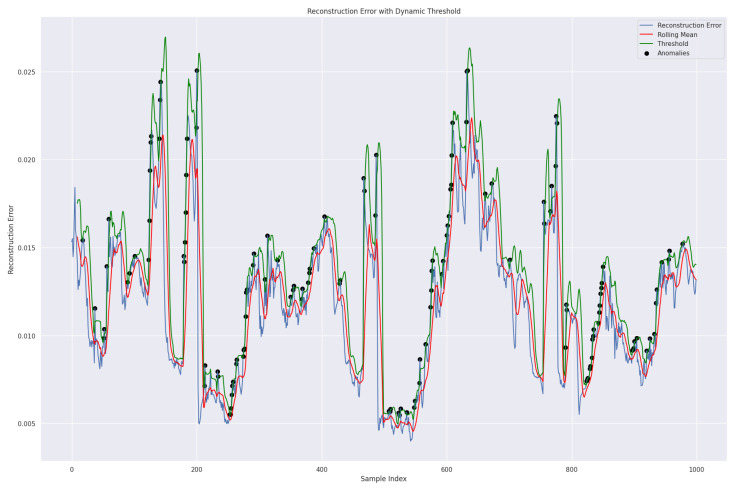
A representation of the proposed TT-TBAD model for anomaly detection with a dynamic threshold based on rolling mean and reconstruction error over a specified window and time for the outdoor fattening barn environment.

**Figure 19 sensors-24-02453-f019:**
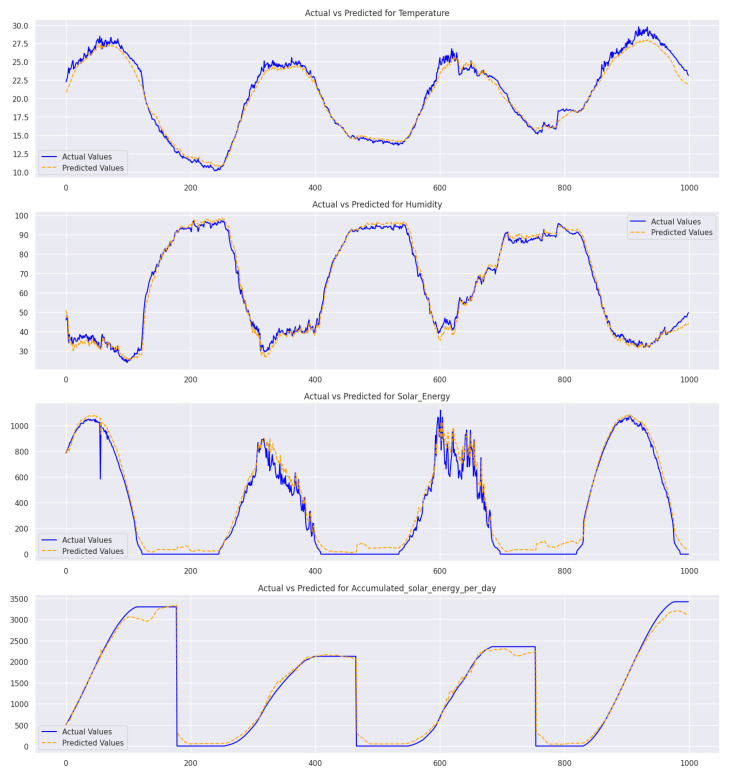
Comparison of actual and predicted values of a small section of the validation set in outdoor weather forecasts in a barn environment for each feature using a multivariate Twin-branch Shared LSTM with Multi-Head Attention.

**Figure 20 sensors-24-02453-f020:**
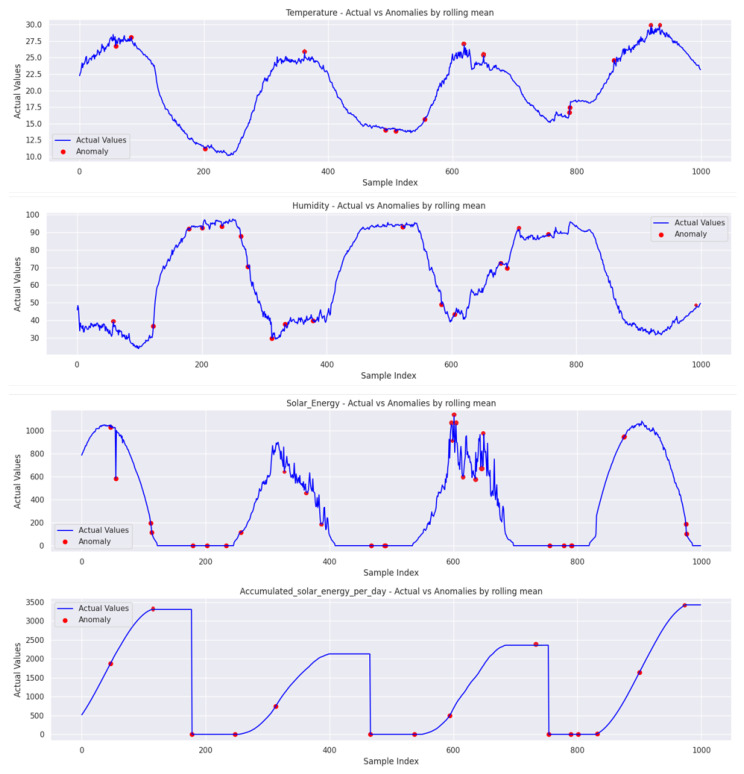
A comparison of the proposed TT-TBAD model for actual and detected anomaly data points in the outdoor fattening barn environment.

**Figure 21 sensors-24-02453-f021:**
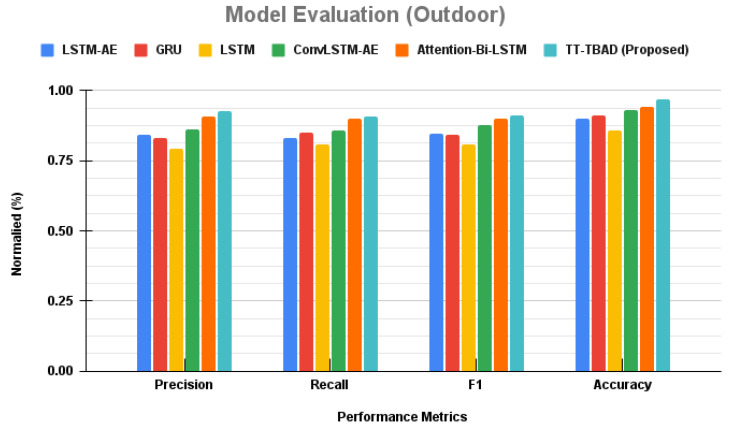
A comparative study of the proposed TT-TBAD for anomaly detection in outdoor environments against various state-of-the-art anomaly detection models.

**Figure 22 sensors-24-02453-f022:**
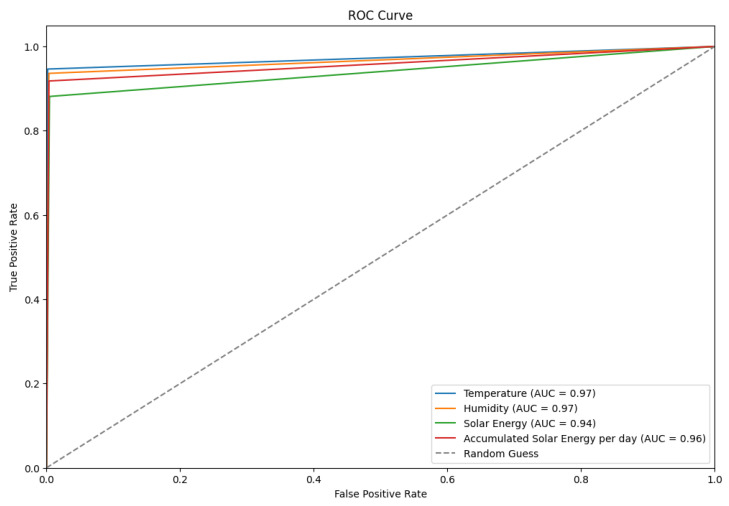
An ROC curve of the proposed TT-TBAD for four classifiers based on environmental parameters of the barn dataset in an outdoor environment.

**Table 1 sensors-24-02453-t001:** A characterization of the various features in the Indoor and Outdoor fattening barn datasets.

**Indoor Fattening Barn**	**Values**
No of sensors	5
No of dataset samples	39,456
No of minutes per instance	5
No of the training set samples	80%
No of test set samples	10%
No of validation set samples	10%
**Outdoor Weather Forecast**	**Values**
No of sensors	4
No of dataset samples	39,456
No of minutes per instance	5
No of the training set samples	80%
No of test set samples	10%
No of validation set samples	10%

**Table 2 sensors-24-02453-t002:** Network architecture for Twin-Branch Shared LSTM with Multi-Head Attention. Bold emphasizes the sections in the table (Input, Attention layers, and Twin branch layers).

Parameter	Values
Input Shape	(24,4), (24,5)
DAE Gaussian Noise	0.1
No. of Shared LSTM Encoder Layers	2
Kernel regularization to each layer with strength	L2(0.001)
** Attention layers **	
• MultiHeadAttention layer	
$\left(d_{model},num_{heads},num_{layers}\right)$	(64,8,0)
• HierarchicalMultiHeadAttention	
$\left(d_{model},num_{heads},num_{layers}\right)$	(64,8,4)
• CrossMultiHeadAttention	
$\left(d_{model},num_{heads},num_{layers}\right)$	(64,8,0)
• FeatureEnchancedMultiHeadAttention	
$\left(d_{model},num_{heads},num_{layers}\right)$	(64,8,4)
**Twin-Branch layers (reconstruction and prediction)**	
Reconstruction Branch (Loss of weight)	(64,1.0)
Prediction Branch (Loss of weight)	(64,0.5)
Optimizer, Loss	Adam, MSE
Batch Size	24,32
Epoch, Callbacks, Feedback Loop	20,4,Yes

**Table 3 sensors-24-02453-t003:** An evaluation of the proposed TT-TBAD model against benchmark models for various features of the indoor fattening barn environment.

Feature	Method	MSE	MAE	RMSE	$R^{2}$ Score
Temperature (°C)	LSTM-AE [21]	0.0023	0.018	0.015	0.84
	GRU [22]	0.00048	0.015	0.026	0.78
	LSTM [22]	0.0062	0.015	0.021	0.79
	ConvLSTM-AE [22]	0.0045	0.0537	0.05	0.81
	Attention-Bi-LSTM [39]	0.0014	0.032	0.038	0.82
	**TT-TBAD (ours)**	**0.00011**	**0.012**	**0.016**	**0.93**
Humidity (%)	LSTM-AE [21]	0.0010	0.032	0.019	0.89
	GRU [22]	0.0007	0.021	0.026	0.91
	LSTM [22]	0.005	0.077	0.026	0.83
	ConvLSTM-AE [22]	0.0032	0.046	0.048	0.86
	Attention-Bi-LSTM [39]	0.0019	0.029	0.033	0.93
	**TT-TBAD (ours)**	**0.0002**	**0.014**	**0.018**	**0.98**
CO_2_ (ppm)	LSTM-AE [21]	0.0011	0.053	0.039	0.71
	GRU [22]	0.001	0.027	0.033	0.67
	LSTM [22]	0.0006	0.01	0.033	0.79
	ConvLSTM-AE [22]	0.0013	0.061	0.048	0.69
	Attention-Bi-LSTM [39]	0.0007	0.021	0.027	0.81
	**TT-TBAD (ours)**	**0.0001**	**0.0098**	**0.013**	**0.94**
NH_3_ (ppm)	LSTM-AE [21]	0.0062	0.053	0.078	0.88
	GRU [22]	0.001	0.032	0.042	0.83
	LSTM [22]	0.001	0.029	0.042	0.85
	ConvLSTM-AE [22]	0.0071	0.061	0.085	0.82
	Attention-Bi-LSTM [39]	0.0019	0.029	0.021	0.91
	**TT-TBAD (ours)**	**0.0011**	**0.024**	**0.013**	**0.94**
H_2_S (ppm)	LSTM-AE [21]	0.071	0.035	0.10	0.11
	GRU [22]	0.081	0.012	0.011	0.16
	LSTM [22]	0.041	0.013	0.15	−0.018
	ConvLSTM-AE [22]	0.027	0.012	0.012	0.12
	Attention-Bi-LSTM [39]	0.00018	0.011	0.013	0.31
	**TT-TBAD (ours)**	**0.00013**	**0.0072**	**0.009**	**0.71**

**Table 4 sensors-24-02453-t004:** A comparative analysis of various anomaly detection models for the indoor fattening barn environment.

Methods	MSE	MAE	RMSE	$R^{2}$ Score
LSTM-AE [21]	0.000112	0.0687	0.00106	0.89
GRU [22]	0.00071	0.0220	0.0086	0.88
LSTM [22]	0.00140	0.0196	0.0533	0.83
ConvLSTM-AE [22]	0.0036	0.0446	0.00367	0.90
Attention-Bi-LSTM [39]	0.00104	0.0147	0.0340	0.97
**TT-TBAD (ours) **	**0.000404**	**0.0137**	**0.000404**	**0.993**

**Table 5 sensors-24-02453-t005:** A comparative analysis of various anomaly detection models for the outdoor fattening barn environment.

Methods	MSE	MAE	RMSE	$R^{2}$ Score
LSTM-AE [21]	0.00311	0.0477	0.0506	0.95
GRU [22]	0.0020	0.0342	0.00202	0.90
LSTM [22]	0.00471	0.0509	0.0638	0.86
ConvLSTM-AE [22]	0.00367	0.0400	0.0491	0.96
Attention-Bi-LSTM [39]	0.0010	0.0197	0.0342	0.97
**TT-TBAD (ours)**	**0.000618**	**0.0120**	**0.00741**	**0.994**

**Table 6 sensors-24-02453-t006:** An evaluation of the proposed TT-TBAD model against benchmark models for various features of the **outdoor** fattening barn environment.

Feature	Method	MSE	MAE	RMSE	$R^{2}$ Score
Temperature (°C)	LSTM-AE [21]	0.0029	0.0415	0.0542	0.88
	GRU [22]	0.00197	0.0372	0.0444	0.91
	LSTM [22]	0.00254	0.04427	0.05045	0.90
	ConvLSTM-AE [22]	0.00306	0.0439	0.05531	0.88
	Attention-Bi-LSTM [39]	0.00075	0.0216	0.0275	0.96
	**TT-TBAD (ours)**	**0.00026**	**0.00418**	**0.00462**	**0.99**
Humidity (%)	LSTM-AE [21]	0.0049	0.0515	0.0705	0.93
	GRU [22]	0.00092	0.0224	0.0303	0.95
	LSTM [22]	0.00174	0.0342	0.0417	0.96
	ConvLSTM-AE [22]	0.00371	0.04481	0.06096	0.94
	Attention-Bi-LSTM [39]	0.0013	0.0290	0.0301	0.97
	**TT-TBAD (ours)**	**0.00045**	**0.0151**	**0.0211**	**0.99**
Solar Energy (w/m^2^)	LSTM-AE [21]	0.0050	0.0490	0.0712	0.72
	GRU [22]	0.00142	0.0241	0.0377	0.86
	LSTM [22]	0.01404	0.0843	0.0102	0.81
	ConvLSTM-AE [22]	0.00231	0.03435	0.04808	0.87
	Attention-Bi-LSTM [39]	0.0014	0.0244	0.0383	0.91
	**TT-TBAD (ours)**	**0.0011**	**0.0201**	**0.0271**	**0.94**
Accumulated Solar Radiation (J/cm^2^)	LSTM-AE [21]	0.0074	0.0489	0.0965	80.0
	GRU [22]	0.00177	0.02591	0.0421	0.93
	LSTM [22]	0.0040	0.0409	0.0638	0.89
	ConvLSTM-AE [22]	0.00561	0.03727	0.07496	0.85
	Attention-Bi-LSTM [39]	0.00172	0.0138	0.0415	0.95
	**TT-TBAD (ours)**	**0.00075**	**0.0084**	**0.0271**	**0.98**

## Data Availability

Due to the proprietary nature of the dataset which serves as a foundational element of our project, we are unable to make it available to the public. We recognize the importance of data transparency and reproducibility in research; however, the confidentiality agreements and the sensitive nature of the data necessitate these restrictions. We are committed to ensuring the integrity of our research within these constraints and are available to answer any queries related to our methodologies and findings, to the extent possible without compromising the confidentiality of the dataset.

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
