# Peer review of "TimeTector: A Twin-Branch Approach for Unsupervised Anomaly Detection in Livestock Sensor Noisy Data (TT-TBAD)"

_sensors, 2024, doi:10.3390/s24082453_

Round 1

Reviewer 1 Report (Previous Reviewer 2)

Comments and Suggestions for Authors

Please find the comments in the attached file.

Comments on the Quality of English Language

Minor editing of English language required.

Author Response

Thank you for critically reviewing our manuscript. We appreciate your insightful comments and the time you have taken to provide feedback. Your suggestions have been invaluable in improving the quality of the manuscript to meet the requirements of the journal. We have taken action as per the reviewer's suggested comments.

Reviewer 2 Report (New Reviewer)

Comments and Suggestions for Authors

1. The definition and implication of the anomaly is unclear. 

The paper's exploration of anomaly detection in time series data lacks the definition and practical implications of detected anomalies. I am confused about the implication of these anomalies on farm operations and whether they represent genuine outliers or simply infrequent observations. This vagueness undermines the practical relevance of the study and contributes to ongoing misconceptions about anomaly detection in time series. The authors are suggested to provide more analysis on this issue. The following paper is recommended:

Wu, R., & Keogh, E. J. (2021). Current time series anomaly detection benchmarks are flawed and are creating the illusion of progress. IEEE transactions on knowledge and data engineering, 35(3), 2421-2429.

2. The presented methodology is confusing. For example, the paper presents 30 equations in total. However, only 3 of them are described.

3. The comparison against the traditional methods seems superficial.

It is evident that with more complex methodologies, higher accuracy is expected. However, the analysis provided is limited. For example, the papers lack a detailed examination of training settings, model complexity, and execution time. A more detailed and rigorous analysis is crucial to validate the claimed superiority and to communicate the potential trade-offs to the reader transparently.

4. I found the paper lacks a discussion of limitations and assumptions.

Comments on the Quality of English Language

The overall presentation is average. Although the paper is readable, the presentation of the English should improved significantly. I noticed a lot of unclear explanation and not-smooth flow of the text. .

Author Response

Thank you for critically reviewing our manuscript. We appreciate your insightful comments and the time you have taken to provide feedback. Your suggestions have been invaluable in improving the quality of the manuscript to meet the requirements of the journal. We have taken action as per the reviewer's suggestions.

Round 2

Reviewer 1 Report (Previous Reviewer 2)

Comments and Suggestions for Authors

The comments have been well handled.

Reviewer 2 Report (New Reviewer)

Comments and Suggestions for Authors

My concerns are well answered.

This manuscript is a resubmission of an earlier submission. The following is a list of the peer review reports and author responses from that submission.

Round 1

Reviewer 1 Report

Comments and Suggestions for Authors

Since there is no ground truth for anomalies, positioning this work as for anomaly detection is highly inappropriate.  The proposed technique appears to do quite well for regression.  I would strongly suggest that the authors present the proposed technique as a regressor.

For anomaly detection, it is critical to know what is being identified as anomaly.  The authors constantly use the term "data" associated with anomaly, which is not appropriate.  Based on the later experiments, the authors appear to be declaring individual recording as anomalies.  This is typically described as point anomaly, see <https://doi.org/10.1145/3444690>.

I don't see how the different types of data values are normalized in this work, but based on the reported values of MSE and so on, the raw values must have been normalized.  Please describe the normalization procedure.  By the way, since some of the values in the thousands while others are in single digits, it is necessary to normalize the different values.

Figure 6 should be moved to the beginning of Section 4.  Section 4 should be restructured to explain the different components of Figure 6.  The symbols appearing in various equations should be explained.  As is, there are too many equations referring to symbols that appear only once and without explanation.  For example, equation (13) seems to be describing some important operations, but I don't see how they are related to any components of Figure 6 nor paragraphs before it. 

The existing list of key contributions appear to be referring to the three components of the proposed TT-TBAD.  If this is the intention of the authors, please make it more clear.

Denoising Autoencoder (DAE) is presented as key contribution 1 on line 94, however, section 4.1 seems to refer the reader to references [32] and [36] , which seems to suggest that the DAE is NOT invented by the authors.  If it is indeed now, please say anything about what's new.

In Tables 1 and 2, there are rows with label "No of instances per min" and answer of "5 minutes," which does not make sense to me.  For the question of "how many number of instances per min," the answer would be either 5 instances per minute or once every 5 minutes (which should be written as 0.2).

Equation (26) probably should be 0.5 < rr < 0.7??

There are a number of sentences running into each other.  Please add space after the end of sentences.

On line 473, the dynamic threshold is defined based on a rolling mean plus k standard deviation, however I can not find more definition of how the rolling mean is computed nor what is k.

In Figure 15, the temperature predictions appear to be constantly less than the actual values.  Is this simply a small section of the overall data values? otherwise, this is a bad prediction.

The results mentioned on line 15 of the abstract of 99.3% and 99.4% are regression measures, not anomaly detection measures.  This should be made clear.

The title and introduction of this work mention "noise" multiple time, but the test results really does not deal with noise in the data in any reasonable ways.  For example, in Figures 17 and 18, the sharp dips in solar energy around index 50 mostly likely should be declared as anomaly or noise, however, the predicted values are following this dip quite precisely.  This should be regarded as an overfitting rather than a good answer.  

Comments on the Quality of English Language

- The 1st sentence of abstract names many different application areas and taking up more than a whole line of text.  This is unnecessary.  Similarly, on line 11, four different versions of multiheadattention mechanisms are mentioned -- I could see that all of them are mentioned explicitly in Figure 6, but is it really necessary to spell them out in the abstract?

- The term "sequential data" on line 5 should be replaced with "timeseries" or something similar.

Author Response

Please find the attached file herewith. 

Reviewer 2 Report

Comments and Suggestions for Authors

Please see the comments in the uploaded file.

Comments on the Quality of English Language

Minor editing of English language required.

Author Response

Please find the attached file herewith

Round 2

Reviewer 1 Report

Comments and Suggestions for Authors

The authors' insist on presenting their technique as anomaly detection, while the actual experiments present quality measures such as RMSE and R2.  These measures are for regression techniques, NOT for anomaly detection / classification techniques.  This is a fundamental misunderstanding that must be addressed.  Claiming their work is on anomaly detection, while stating "results of our experiments are 99.3% and 99.4%" suggests to me that the classification results were 99.3% and 99.4% accurate.  However, this is not the case.  Based on what I could find out, these two numbers that were stated as from Table 5, are actually R2 scores, not accuracy numbers.  Furthermore, table 5 does NOT actually contain any numbers that are 99.3% (or equal to 99.3%) -- the closest entries I could identify are .99.  Since these two numbers are critical to the overall strength of the work, I would be essential for the authors to make sure they are well documented and do not require the readers to guess about them.

Yes, there are other people who use regression techniques to solve anomaly detection problems.  The problem with the current work is that it improves upon the modeling for normal events of the anomaly detection, not the overall anomaly detection procedure.  The intellectual honest way to state the contribution would be say the authors are proposing a more accurate model building procedure.

Following up on what authors identified as "comment 13" from the original review, could the authors please be more direct about how their work address the "noise" in the raw data?  Yes, DAE was meant to deal with noisy data, but what are the contribution from this work that address the noise present in the data?  Since the model built by TT-TBAD agrees extremely well with the input data (with R2 = 0.993 and 0.994), it seems to this reviewer that the model is capture considerable portion of the "noise" in the raw input data.  Is it possible to quantify how the DAE or TT-TBAD better handle the noise?

Comments on the Quality of English Language

On line 4, "The research centers..." requires some revision to avoid confusion.  "The research" presumably is "This research" because I don't see a reasonable target "the research" would be referring to.  On my 1st reading of this sentence, I was assuming "the research centers" was a noun phrase, and has to read till the end of the sentence to figure out that is no verb in the sentence before I come back to re-parse the word "centers" as a verb.  It would be good to choose a different word here to avoid such confusion.

On line 15, please choose a more precise word to replace the word "results" in the phrase "the results of our experiments."

Author Response

All comments have been answered carefully. Please see the attachment.

Reviewer 2 Report

Comments and Suggestions for Authors

The comments have been handled.

Comments on the Quality of English Language

Minor editing of English language required.

Author Response

All the comments have been answered carefully. Please see the attachment.
